# The Laser Selective Sintering Controlled Forming of Flexible TPMS Structures

**DOI:** 10.3390/ma16247565

**Published:** 2023-12-08

**Authors:** Chenhao Xue, Nan Li, Shenggui Chen, Jiahua Liang, Wurikaixi Aiyiti

**Affiliations:** 1School of Mechanical Engineering, Xinjiang University, Urumqi 830047, China; xch1107@stu.xju.edu.cn (C.X.); dglgln@163.com (N.L.); 2School of Education (Normal School), Dongguan University of Technology, Dongguan 523808, China; 3School of Art and Design, Guangzhou Panyu Polytechnic, Guangzhou 511483, China; 4Dongguan Institute of Science and Technology Innovation, Dongguan University of Technology, Dongguan 523808, China; jiahua_liang1114@163.com

**Keywords:** metamaterials, TPMS hybrid gradient structure, mechanical response of flexible lattice structure, laser selective sintering, thermoplastic polyurethane (TPU)

## Abstract

Sports equipment crafted from flexible mechanical metamaterials offers advantages due to its lightweight, comfort, and energy absorption, enhancing athletes’ well-being and optimizing their competitive performance. The utilization of metamaterials in sports gear like insoles, protective equipment, and helmets has garnered increasing attention. In comparison to traditional truss and honeycomb metamaterials, the triply periodic minimal surface lattice structure stands out due to its parametric design capabilities, enabling controllable performance. Furthermore, the use of flexible materials empowers this structure to endure significant deformation while boasting a higher energy absorption capacity. Consequently, this study first introduces a parametric method based on the modeling equation of the triply periodic minimal surface structure and homogenization theory simulation. Experimental results demonstrate the efficacy of this method in designing triply periodic minimal surface lattice structures with a controllable and adjustable elastic modulus. Subsequently, the uniform flexible triply periodic minimal surface lattice structure is fabricated using laser selective sintering thermoplastic polyurethane technology. Compression tests and finite element simulations analyze the hyperelastic response characteristics, including the element type, deformation behavior, elastic modulus, and energy absorption performance, elucidating the stress–strain curve of the flexible lattice structure. Upon analyzing the compressive mechanical properties of the uniform flexible triply periodic minimal surface structure, it is evident that the structure’s geometric shape and volume fraction predominantly influence its mechanical properties. Consequently, we delve into the advantages of gradient and hybrid lattice structure designs concerning their elasticity, energy absorption, and shock absorption.

## 1. Introduction

In recent years, the focus on sports equipment has shifted towards advancements in material science and structural design. Among the innovations in mechanical metamaterials, the flexible porous structure stands out. This structure has gained significant attention due to its remarkable qualities: high elasticity, excellent energy absorption, and shock absorption. Not only does it lighten the sports equipment, but it also helps lower the risk of athletes getting injured. The design of porous structures now plays a pivotal role in revolutionizing sports gear [1,2,3]. In comparison to traditional lattice structures like face-centered cubic, body-centered cubic, and honeycomb structures, the triply periodic minimal surface (TPMS) lattice structure has gained prominence. Its appeal lies in its being lighter and possessing superior energy-absorbing and shock-absorbing traits [4]. Nonetheless, producing these structures has long been a challenge using conventional methods.

The advancement of additive manufacturing has made it possible to use lattice structures in sports equipment [5]. Laser selective sintering, with its rapid formation and having no requirement to remove support parts, has emerged as an ideal method for creating lattice structures. As material science progresses, flexible materials capable of enduring significant elastic deformation are being utilized in additive manufacturing. These materials, replacing traditional foam, offer cushioning functions in personalized sports equipment tailored to the diverse needs of athletes [6]. For instance, in healthcare, personalized, flexible lattice structures offer individualized support for clinical patients, aiding in the treatment or relief of conditions like pressure sores [7]. Similarly, flexible lattice structure insoles have been developed to address movement issues during activities such as walking, standing, or running [8]. The potential of additive manufacturing in creating flexible metamaterials holds great promise for producing protective gear for sports equipment and personalized medical aids [9].

The TPMS structure offers a key advantage through its parametric design, which is based on implicit function equations. These design parameters—like element size, wall thickness, and arrangement—directly impact the mechanical properties of lattice structures [10,11]. A lot of research has delved into the mechanical properties, deformation behavior, and structural optimization of TPMS lattice structures. Recent studies have particularly focused on rigid lattice structures, such as the Gyroid and Diamond, crafted from robust materials like polymers (for instance, PA12) or metals (like Ti-6Al-4V) [12,13,14,15]. These studies mainly highlight how the wall thickness affects properties such as the elastic modulus, energy absorption, fracture stress, and deformation behavior [16,17,18]. However, in applying lattice structures in engineering, the design optimization must align with the functional requirements of the parts [19]. Although there have been proposed optimization methods for gradient and hybrid TPMS lattice structures [20], considering that most lattice structure preparation involves support, further improvement is needed in researching the preparation and performance of these structures [21,22]. In recent years, the advancement of laser selective sintering polymer material technology has broadened the possibilities for unsupported printing, allowing the creation of various lattice structures. Consequently, numerous rigid gradient and hybrid TPMS lattice structures have been manufactured to assess their mechanical properties [23,24,25]. Studies indicate that compared to uniform lattice structures, rigid gradient and hybrid lattice structures made from tough materials can enhance energy absorption and efficiency [26,27]. Yet there is a scarcity of research simulating and characterizing the mechanical properties of flexible lattice structures [28,29,30].

This study introduces a novel method to design a flexible TPMS lattice structure by connecting the geometric parameters of the TPMS structure with its mechanical properties through simulation. We utilized selective laser sintering technology to create uniform, gradient, and hybrid lattice structures. To validate the effectiveness of our design method, compression tests and structural finite element analyses were conducted. Additionally, we analyzed the hyperelastic mechanical behavior of the flexible lattice structure. Our findings show that the experimental results demonstrate the efficacy of this method in designing triply periodic minimal surface lattice structures with controllable and adjustable elastic moduli, and optimizing the design using gradient and hybrid techniques significantly enhanced the mechanical properties of the flexible TPMS lattice structure.

## 2. Materials and Methods

### 2.1. Parameterized Structural Design

#### 2.1.1. TPMS Structural Design

To provide a clearer understanding of the mathematical definition of TPMS structures, the following definitions are introduced. First, in the Cartesian coordinate system, the structure’s period, denoted as *k*, is defined as follows:(1)$$k_{i}=2\pi n_{i}$$

In this context, *i* represents the x, y, and z directions, while $n_{i}$ stands for the number of unit repetitions of the TPMS lattice structure in the *i* direction. Next, let us define the shorthand symbols for the sine and cosine periodic functions as follows:(2)$$S_{i}=\sin\left(k_{i}\frac{i}{L_{i}}\right)$$
and
(3)$$C_{i}=\cos\left(k_{i}\frac{i}{L_{i}}\right)$$

$L_{i}$ represents the length of one unit within a lattice structure in the *i* direction. The function of the Gyroid ($U_{G}$) and Diamond ($U_{D}$) structures can be defined as follows:(4)$$U_{G}\left(x,\;y,\;z\right)=S_{x}C_{y}+S_{y}C_{z}+S_{z}C_{x}-T$$
(5)$$U_{D}\left(x,\;y,\;z\right)=C_{x}C_{y}C_{z}+S_{x}S_{y}C_{z}+S_{y}S_{z}C_{x}+S_{z}S_{x}C_{y}-T$$

*T* represents an adjustable parameter that regulates the volume fraction, or wall thickness, which indirectly influences the structure of TPMS. The volume fraction (*V*) is defined as follows:(6)$$V=\frac{V_{wall}}{V_{wall}+V_{void}}$$

The $V_{wall}$ and $V_{void}$ indicate the volumes of the solid material and the empty spaces within the lattice structure, respectively.

#### 2.1.2. Gradient and Hybrid TPMS Structure Design

By substituting a new function, $T\left(x,y,z\right)$, for the original fixed value, *T*, into Equations (4) and (5) describing the TPMS structure, we can bring about a gradient change in the volume fraction along a specified direction within the lattice structure arrangement. This function is defined as follows:(7)$$\rho_{G}\left(x,\;y,\;z\right)=S_{x}C_{y}+S_{y}C_{z}+S_{z}C_{x}-T\left(x,\;y,\;z\right)$$
(8)$$\rho_{D}\left(x,\;y,\;z\right)=C_{x}C_{y}C_{z}+S_{x}S_{y}C_{z}+S_{y}S_{z}C_{x}+S_{z}S_{x}C_{y}-T\left(x,\;y,\;z\right)$$

The functions $\rho_{G}\left(x,y,z\right)$ and $\rho_{D}\left(x,y,z\right)$ represent the Gradient Gyroid and Diamond structures, respectively. The function $T\left(x,y,z\right)$ defines the change in the volume fraction gradient within the lattice structure’s dimensions along the x, y, and z axes. The values within the range of *T* correspond to the mapped volume fraction gradient changes.

When two implicit function equations create a lattice structure where one type of element’s gradient transitions to another, it is termed a ‘hybrid structure’. At the intersection of these two lattice forms, a continuous function can smoothly connect the morphologies. Consequently, the entire structure can be described as follows:(9)$$H_{U}\left(x,\;y,\;z\right)=a\left(x,\;y,\;z\right)U_{1}\left(x,\;y,\;z\right)+\left(1-a\left(x,\;y,\;z\right)\right)U_{2}\left(x,\;y,\;z\right)$$

$H_{U}$, $U_{1}$, and $U_{2}$ denote the functions associated with the resultant hybrid structure, the initial parent unit structure 1, and the initial parent unit structure 2, respectively. The weight function is denoted as $\alpha \left(x,y,z\right)\in \left[0,1\right]$. The sigma function (SF) is employed in this research as a weight function to depict the gradual transition from 0 to 1. It is defined as follows:(10)$$a\left(x,\;y,\;z\right)=\frac{1}{1+e^{-KG\left(x,\;y,\;z\right)}}$$

The equation $G\left(x,y,z\right)$ = 0 represents the border at which the morphology of the two elements overlaps. This equation defines a continuous function $G\left(x,y,z\right)$. The parameter *K* (where *K* > 0) can be utilized to modify the transition gradient.

### 2.2. Parametric Controlled Forming

As shown in Figure 1, firstly, the mathematical relationship $f\left(T,V\right)$ between the volume fraction *V* and fixed curvature *T* can be fitted from the different values of Equations (4)–(6). At a given volume fraction, the homogenization theory can simulate the elastic modulus of the TPMS unit structure. Then the relationship $f\left(V,E\right)$ between the volume fraction *V* and the elastic modulus *E* is determined by the nonlinear fitting. The parametric design of a flexible TPMS lattice structure with controllable performance can be realized by combining $f\left(T,V\right)$ and $f\left(V,E\right)$.

The experiment showed that for unit sizes larger than 10 mm, having a volume fraction between 15% and 30% works best for laser selection sintering forming. To achieve this, we set the TPMS structural unit size to 12 mm and the period k to 3 × 3 × 3 using Mathematica. We created three homogeneous TPMS lattice structures with volume fractions of 15%, 20%, and 25%, respectively. We used TPM3D P360 (TPM3D, Shanghai, China) laser selective sintering equipment to prepare the samples using flexible thermoplastic polyurethane (TPU, WANHUA, Shanghai, China) as the material. Table 1 and Table 2 summarize the key processing parameters and the quality assessment of the printed samples. (Process parameters are recommended by equipment and material suppliers).

### 2.3. Quasi-Static Compression Experiment of TPMS Lattice Structure

The mechanical properties and energy absorption of the scaffolds made through selective laser sintering were assessed via compression experiments. A Shanghai Force Test LD23.104 10 KN microcomputer-controlled electronic universal testing machine (from China Force Test Scientific Instrument Co., Shanghai, China) was used, following the international standard for testing polymer materials: BS EN ISO 3386-1:1997+A1:2010 [31]. Prior to testing, samples were conditioned at 23 °C and 50% relative humidity for at least 16 h. Compression was applied at a strain rate of 5 mm/min and a strain of 0.75, with the corresponding compression-displacement data recorded over three consecutive load-to-unload cycles.

Figure 2 illustrates the mechanical response curves of both a general lattice structure [32] and a TPMS lattice structure finite element model (Figure 2b). Typically, when a lattice structure undergoes compressive loading, it experiences three stress–strain behavior stages: elastic, plateau, and densification (as depicted in Figure 2 [32]). Initially, during compression, the lattice structure enters the elastic stage where the stress–strain relationship is linear, defined by the elastic modulus *E* as follows:(11)E=σ/ε

Afterward, it progresses to the platform stage, where the stress on the platform remains mostly steady, while the strain keeps rising until it reaches the densification stage. In this stage, the stress suddenly escalates until it hits the highest point. It is during this phase that we can describe the energy absorption traits using two mathematical measures: the amount of energy absorbed, denoted as W, and the energy dissipation difference (Δ*W*) between loading and unloading.
(12)$$W=\int_{0}^{\varepsilon }\sigma \left(\varepsilon \right)d\varepsilon$$
(13)$$\Delta W=W_{load}-W_{Unload}$$

### 2.4. Finite Element Analysis of TPMS Structure

The flexible TPMS structural unit’s elastic properties were determined using Dong et al.’s three-dimensional finite element homogenization method [33]. This method calculates the homogenization stiffness matrix, CH, for a specific element. Dong et al. shared a MATLAB code named Homo3D for this method, which takes the first and second Lamé parameters of the lattice’s constitutive material as input. The lattice structural unit is represented as a voxel model.

Nguyen et al. previously used the Homo3D program to obtain effective material properties for an additive manufacturing lattice structure [34]. They found that it accurately predicts the elastic load-deflection response in a three-point bending experiment.

The formula below provides the first and second Lamé parameters:(14)$$\lambda =\frac{\nu E}{\left(1+v\right)\left(1-2v\right)}$$
and
(15)$$\mu =\frac{E}{2\left(1+v\right)}$$

Here, *E* represents the modulus of elasticity of the lattice’s constitutive material, and v is Poisson’s ratio. For this study, Poisson’s ratio and the elastic modulus were set at 0.38 and 14 MPa, respectively. These values are typical for the tensile and compressive moduli of TPU manufactured using the same laser selective sintering machine and particles [35].

The stiffness matrix $C^{H}$ can be obtained from the Homo3D code. Its inverse matrix, $\;S{\;}^{H}$, represents the flexibility matrix of the structure. This allows for an effective representation of the lattice structure’s elastic properties through a three-dimensional modulus surface. The elastic modulus along the [100], [010], and [001] directions is defined as follows:(16)$$E_{\left[100\right]}=\frac{1}{S_{11}^{H}}$$
(17)$$E_{\left[010\right]}=\frac{1}{S_{22}^{H}}$$
(18)$$E_{\left[001\right]}=\frac{1}{S_{33}^{H}}$$

In order to evaluate the elastic anisotropy of the same structure under the same volume fraction, we use the difference between the maximum elastic modulus and the minimum elastic modulus obtained by the three-dimensional modulus surface to evaluate the difference range of its elastic modulus, which is mathematically defined as follows:(19)$$\Delta E=E_{max}-E_{min}$$

We conducted a three-dimensional finite element (FE) analysis to study how flexible TPMS lattice structures respond to stress and strain. In our compression test model (Figure 2b), two rigid plates mimic the indenter and bottom plate. We used solid mechanics and a steady-state analysis, fixing the bottom plate as a boundary condition while applying forced displacement for loading. Due to the TPMS structure’s complexity and the TPU material’s hyperelasticity, we employed the Neo-Hookean superelastic constitutive model to predict the TPMS structure’s deformation. This model, commonly used for hyperelastic materials, is straightforward and effective in predicting deformation patterns. In COMSOL, we utilized the Lamé parameter to build the Neo-Hookean hyperelastic model, defined mathematically as:(20)$$W=C_{1}\left(I_{1}-3\right)$$

Here, C_1_ (C_1_ = μ/2) represents the material constant, I_1_ signifies the strain tensor, and μ is defined by Equation (15).

## 3. Results

### 3.1. Parametric Design Model

The curvature, *T*, of each type of TPMS structure determines a unique function for the volume fraction, *V*. Different TPMS lattice structures can be obtained by plugging a range of *T* values into Equations (4) and (5). Equation (6) allows for the calculation of the volume fraction, *V*, corresponding to each curvature, *T*, value. Finally, the relationship between the curvature and volume fraction, $f\left(T,V\right)$, of the two designed TPMS structural units (Figure 3) can be obtained through linear regression.
(21)$$f{\left(T,\;V\right)}_{Gyroid}:\;T=1.154\;V+0.0126$$
(22)$$f{\left(T,\;V\right)}_{Daimond}:\;T=1.194\;V+0.0111$$

Figure 4 displays the three-dimensional elastic modulus analysis of two uniform TPMS structural units with varying volume fractions (15%, 20%, 25%, and 30%), following the method outlined in Section 2.4. Firstly, different TPMS structural units exhibit distinct anisotropy ranges. The Gyroid structure’s elastic modulus surface closely resembles that of a sphere, indicating its nearly isotropic properties [36] and similar elastic moduli in various directions. Meanwhile, the Diamond structure becomes more spherical at a 30% volume fraction. Secondly, the influence of the volume fraction on the anisotropy range remains consistent across different TPMS structural elements. The elastic modulus range of both the Gyroid and Diamond structures gradually increases with higher volume fractions. Moreover, both structural units showcase identical elastic moduli in the [001], [100], and [010] directions (as illustrated in Figure 4). In particular, Figure 4a highlights that the three-dimensional elastic modulus of the Gyroid structure surface in the [001], [100], and [010] directions is equal to and lower than that in other directions. On the contrary, Figure 4b demonstrates that the Diamond structure’s elastic modulus in the [001], [100], and [010] directions surpasses that in other directions. Recognizing that TPMS structures with volume fractions of 10% to 30% are not entirely isotropic [37], considering that the two TPMS structures with a volume fraction of 10–30% are not completely isotropic [37], we selected the homogenization simulation results in the directions of [001], [100], and [010] to carry out the nonlinear fitting of the elastic modulus and volume fraction (refer to Figure 3). The mathematical relationship, $f\left(V,E\right)$, is as follows:(23)$$f{\left(V,\;E\right)}_{Gyroid}:\;E=18.54V^{1.2}$$
(24)$$f{\left(V,\;E\right)}_{Daimond}:\;E=20.77V^{1.14}$$

### 3.2. Uniform Flexible TPMS Lattice Structure

The stress–strain curves for two types of similar TPMS lattice structures, each having volume fractions of 15%, 20%, and 25%, were determined through compression tests. Figure 5 displays the stress–strain curves for various lattice structures during the first three cycles of loading and unloading at a compression of 0.75 strain. Observations from these tests indicate some interesting behaviors. First, these flexible lattice structures exhibit the Mullins effect during cyclic compression experiments. This effect means that the slope of the stress–strain curve under the first load is higher than that under subsequent loads, eventually stabilizing over subsequent cycles. Additionally, these structures show hysteresis in their load–unloading stress–strain curves. What this means is that during the same cyclic loading and unloading, the unloading stress–strain curve does not follow the same path as the loading curve. This characteristic implies energy dissipation is occurring within the flexible TPMS lattice structure during cyclic compression. As a result, part of the mechanical energy gets converted into the structure’s internal energy, leading to less work being done during unloading compared to loading [38].

To delve deeper into their deformation behavior, a finite element structural analysis was conducted. For instance, considering the Gyroid and Diamond structures with a 20% volume fraction, as shown in Figure 6, the initial loading caused the cell wall to bend and elastically deform, corresponding to the elastic stage in the stress–strain curve. At a strain of 0.25, deformation was evident in all pore walls. As strain increased further, upon reaching the platform stage, the pores within the cell wall began collapsing, resulting in an increased relative density of the structure. Around a strain of 0.50, most pore walls had collapsed, and upon entering the densification stage at a strain of 0.75, the inner pores of the cell wall completely collapsed and compacted under the load. This led to a sharp increase in stress, and the mechanical properties gradually approached those of the base material. Importantly, after cyclic loading and unloading at 0.75 strain compression, all flexible lattice structures remained intact without any cell wall ruptures or permanent deformations. Hence, all subsequent analyses of the compressive mechanical properties in this study will focus on the characteristics observed in the third loading curve.

The mechanical properties of the various lattice structures are summarized in Table 3, based on their stress–strain curves. These properties include the elastic modulus (*E*), energy absorption (*W*), and energy dissipation (Δ*W*), which signifies the difference between the loading and unloading work. Through the mathematical analysis in Section 3.1, using a nonlinear fitting approach, we derived a relationship denoted as $f\left(V,E\right)$. When the volume fraction is 0.15, 0.20, and 0.25, we obtained elastic modulus values for the Gyroid and Diamond cell structures. For example, the Gyroid structure showed modulus values of 1.92, 2.63, and 3.49 MPa, while the Diamond structure showed 2.42, 3.24, and 4.24 MPa, respectively. Comparing these values with the experimental data, we found an error within 15%. This confirms that our fitted nonlinear relationship aligns well with experimental findings, offering guidance for controlling the mechanical properties of flexible TPMS structures. Additionally, we observed that as the volume fraction increases, both the elastic modulus and the energy absorption of all lattice structures increase, aligning with previous research on rigid lattice structures [39]. For instance, the Gyroid structure’s elastic modulus rose from 2.28 MPa to 3.44 MPa, and the Diamond structure’s rose from 2.60 MPa to 4.43 MPa. This increase can be attributed to thicker walls or an increased mass within the lattice structure, enhancing its mechanical properties [40]. Furthermore, distinct differences in the elastic modulus among various flexible TPMS structures were evident. For the same volume fractions, the Diamond structure consistently exhibited higher rigidity compared to the Gyroid structure. This implies that, at equal volume and mass, the Diamond structure’s geometry provides greater rigidity and energy absorption under stress, and this difference amplifies with increasing volume fraction. Finally, our evaluation of the energy dissipation in all flexible TPMS lattice structures revealed that Diamond structures consistently exhibit higher dissipation than Gyroid structures. In engineering applications, structures with greater energy dissipation tend to have better durability under cyclic loading and improved seismic performance. These findings highlight how volume fraction affects the elastic modulus and energy absorption of Gyroid and Diamond structures in compression experiments, showcasing the superior rigidity and energy absorption capacity of Diamond structures.

### 3.3. Flexible Gradient and Hybrid TPMS Lattice Structure

After analyzing the compressive mechanical properties of the uniform flexible TPMS structures, we found that the structure’s shape and the volume fraction it takes up are the main characteristics that affect how strong it is. So, we looked at two ways to design these flexible structures: one where the volume fraction changes gradually and another where different shapes mix together. In Figure 7a, we made a flexible hybrid and gradient TPMS structure, following the steps in Section 2.1.2. To compare this with what we studied earlier, we kept the total space taken up the same for all the structures at 0.2, and they were arranged in the same way. For the gradient structure in Figure 7b, the change happens along the z-axis, ranging from 0.15 to 0.25. In a hybrid structure, we used the function $KG\left(x,y,z\right)$ *=*
z to transition between different shapes, as seen in Figure 7c. The hybrid structure has three layers: the first is a Gyroid, the second is a smooth connected coating, and the third is a Diamond. Once we finished designing, we used the laser selective sintering process and materials described in Section 2.2. Then, we tested how strong they were using the same compression experiment combined with computer analysis.

Figure 8 displays the stress–strain patterns of two types of structures—the flexible gradient and hybrid TPMS lattice—during the third cyclic loading. When we examine their compressive behavior, both the gradient and hybrid structures exhibit elastic, plateau, and densification stages [41]. In Figure 9, comparing how these structures deform under different strains using finite element simulation and experimentation, we notice differences between the gradient/hybrid flexible TPMS and the uniform lattice structure. The TPMS structure with a volume fraction gradient always starts deforming from the area with the lowest volume fraction and then collapses layer by layer toward regions with higher volume fractions. As for the hybrid structure, the deformation initiates in the Gyroid part, spreads to the transition zone, and concludes in the Diamond part.

The stress–strain curves provide insight into the mechanical properties. Initially, for the Gyroid and Diamond structures exhibiting a continuous linear gradient change in volume fraction from 0.15 to 0.25, their elastic moduli are 2.66 and 3.51 MPa, respectively. Comparing the stress–strain curves of different volume fractions within the same structural element reveals that the elastic modulus of the gradient structure is close to, yet higher than, that of the lowest volume fraction part, a fact confirmed by its deformation behavior.

For the hybrid structure, an elastic modulus of 2.56 MPa lies between the Gyroid and Diamond structures with identical volume fractions, suggesting that the transition region insignificantly weakens its elastic properties. Keeping the volume fraction constant, the Gyroid structure’s elastic modulus can be enhanced by 12% through a hybrid structure design.

Moreover, energy absorption and dissipation are crucial mechanical properties of interest. When compared in the Gyroid and Diamond structures with a constant volume fraction of 0.2, the flexible gradient Gyroid and Diamond structures show increased energy absorption and dissipation, measuring at 1.26, 0.43, and 1.61, 0.52 $\mathrm{MJ}/m^{3}$, respectively. Compared to the Gyroid structure with a constant volume fraction of 0.2, the Gyroid and Diamond hybrid structures exhibit increased energy absorptions of 1.19 and dissipations of 0.43 $\mathrm{MJ}/m^{3}$, respectively

## 4. Discussion

The elastic properties of TPMS structures are influenced by geometry and volume fraction. Both the Gyroid and Diamond structures exhibit anisotropy, with the overall anisotropy of the Gyroid structure being lower than that of the Diamond structure. The anisotropy ranges of all TPMS structural units vary, and the elastic modulus ranges gradually increase with a rise in volume fraction. The TPMS structural element shares the same elastic modulus in the [001], [100], and [010] directions due to geometric symmetry. However, the three-dimensional elastic modulus of the Gyroid structure in these directions is identical and lower than that in other directions. Conversely, the elastic modulus of the Diamond structure in the [001], [100], and [010] directions is higher compared to that of other directions, indicating isotropic properties in these three directions.

The stress–strain curve’s hyperelastic response in the flexible TPMS structure exhibits characteristics such as the hysteresis phenomenon and capacity dissipation, demonstrating a nonlinear relationship. However, the stress–strain curves from the initial loading inadequately capture the mechanical properties of the flexible TPMS lattice structures. Notably, the curve displays its steepest slope during the first loading, subsequently stabilizing with increased loading cycles, showcasing a significant disparity between these phases. Consequently, in compression tests for flexible TPMS lattice structures, it is crucial to consider the number of cycles required for the stress–strain curve to reach a stable state rather than relying on the value from the initial load. This is especially relevant concerning the deformation of flexible TPMS structures.

Moreover, the variance in the geometry and volume fraction within TPMS structures primarily influences the divergence in mechanical properties. Hence, the proposed design approach for flexible gradient and hybrid TPMS structures, while maintaining the same element type (Table 4), compares the energy absorption and dissipation properties between the flexible gradient Gyroid and Diamond structures having identical volumes and masses. The observed increases were 22.3% and 16.2% for energy absorption and 19.3% and 18.2% for energy dissipation in the Gyroid and Diamond structures, respectively. Additionally, the hybrid Gyroid and Diamond structures exhibited improvements of 15.5% and 16.2% in both energy absorption and dissipation, respectively.

## 5. Conclusions

This research focuses on controlling the shape of a flexible TPU metamaterial structure using parameters. We developed equations $f\left(V,E\right)$ and $f\left(T,V\right)$ to link the structure, shape, and flexibility of the TPMS lattice. These equations match the experimental results within 15%, showing a strong match between theory and practice. They can guide how we shape flexible TPMS structures, adjusting their properties like elasticity and energy absorption [42,43,44].

We also studied how the flexible TPMS lattice behaves when compressed. Our tests revealed the Mullins effect, hysteresis, and energy dissipation. This structure shows promise in absorbing energy and resisting shocks. The shape and amount of material greatly impact its properties. By changing these factors while keeping the size and weight constant, we can enhance the elasticity, energy absorption, and dissipation. This could improve the comfort of sports and medical gear [45,46].

These enhancements are crucial to the sports and medical fields. Increased elasticity offers better support, especially in medical tools like orthotics and stents. Energy absorption and dissipation are vital for protective gear, shoes, and rehabilitation equipment. Flexible TPMS structures efficiently absorb energy from stress or impact, reducing bounce and vibration [45,46]. This is key for joint and muscle health, intense workouts, and medical treatments [47].

In summary, improving the elasticity, energy absorption, and dissipation of flexible TPMS structures benefits the sports and medical fields [47]. Their customizable nature suits various applications, ensuring better performance, comfort, and safety [47].

## Figures and Tables

**Figure 1 materials-16-07565-f001:**
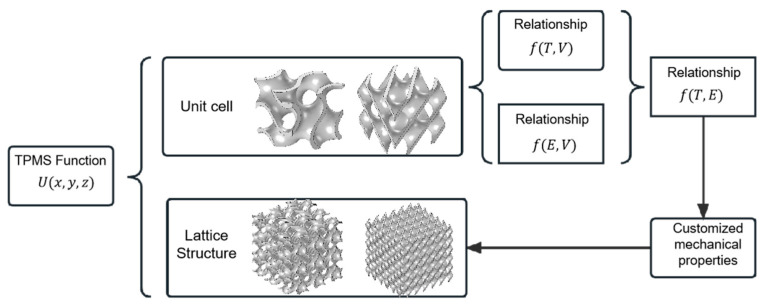
Parameterized TPMS lattice structure design method with a controllable and adjustable elastic modulus.

**Figure 2 materials-16-07565-f002:**
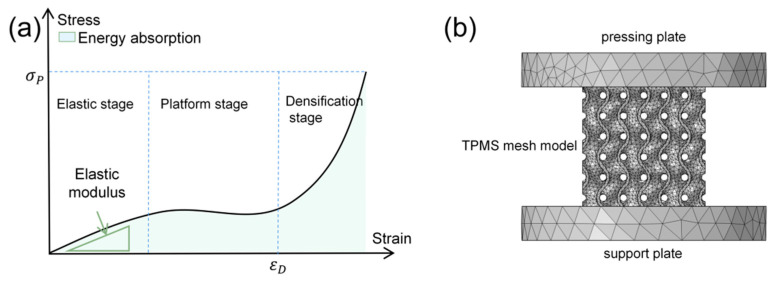
(**a**) Mechanical response curve of general lattice structure [32]; (**b**) TPMS lattice structure finite element model.

**Figure 3 materials-16-07565-f003:**
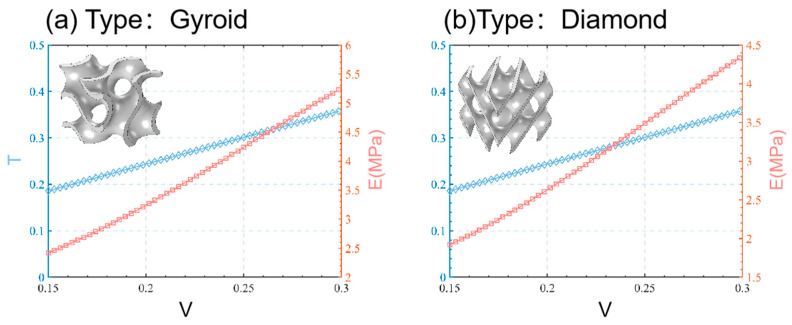
The relationship between geometric parameters and the elastic modulus of a uniform TPMS unit structure.

**Figure 4 materials-16-07565-f004:**
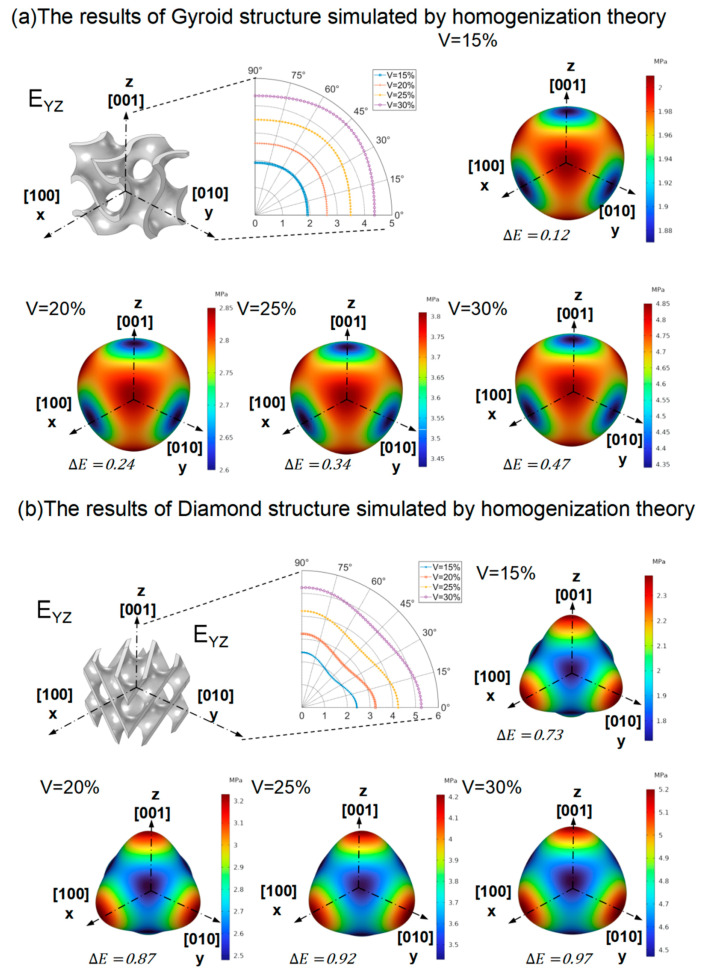
Homogenization theory yields three-dimensional elastic modulus surfaces of Gyroid and Diamond structures.

**Figure 5 materials-16-07565-f005:**
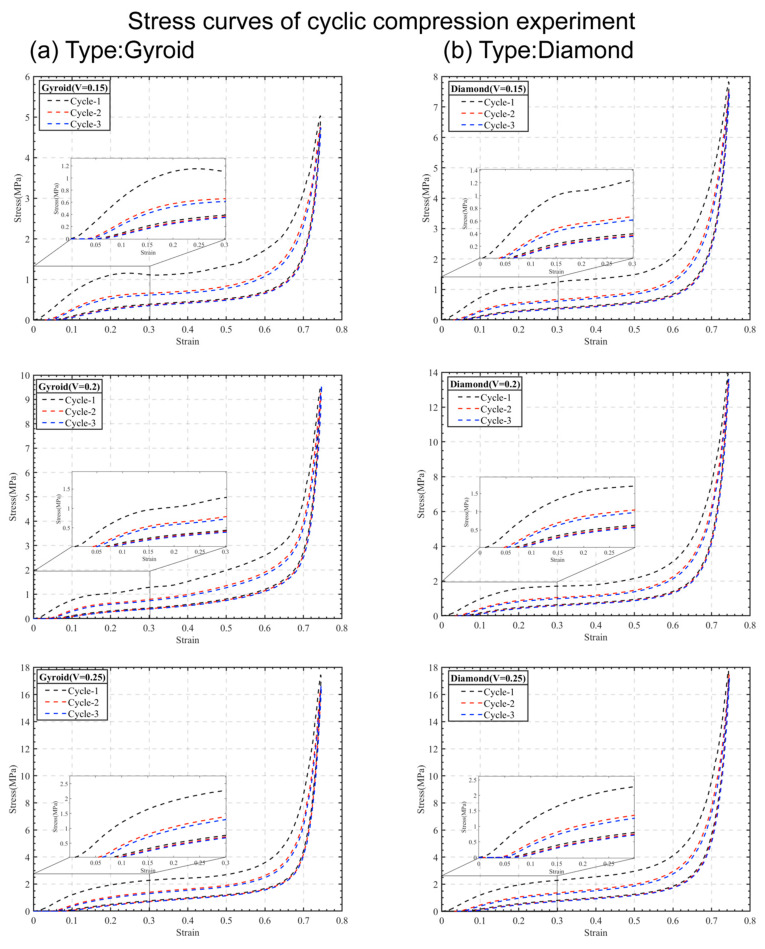
The stress–strain contours of Gyroid and Diamond lattice structures with volume fractions of 0.15 to 0.25.

**Figure 6 materials-16-07565-f006:**
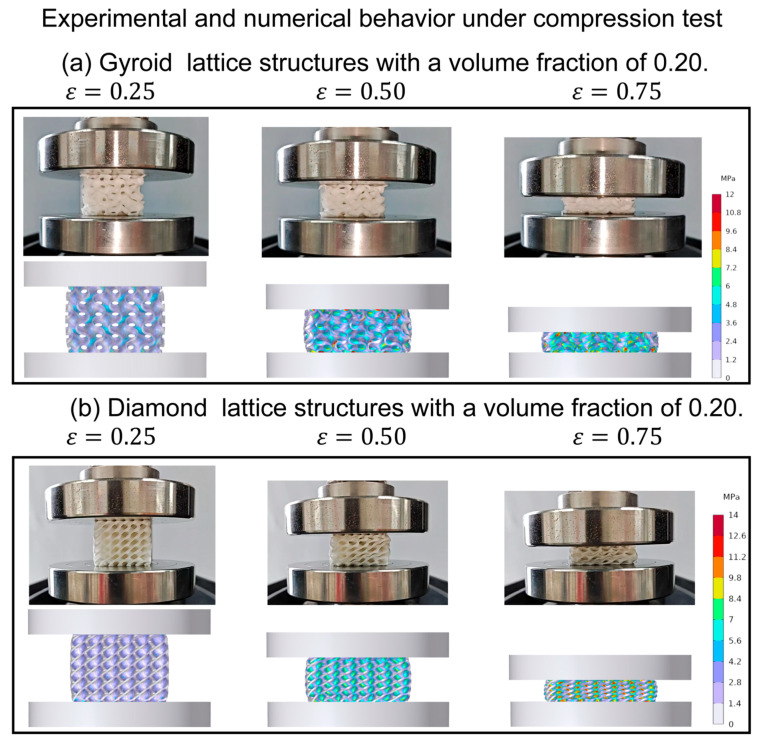
Compressive load finite element deformation analysis of Gyroid and Diamond lattice structures with a volume fraction of 0.20.

**Figure 7 materials-16-07565-f007:**
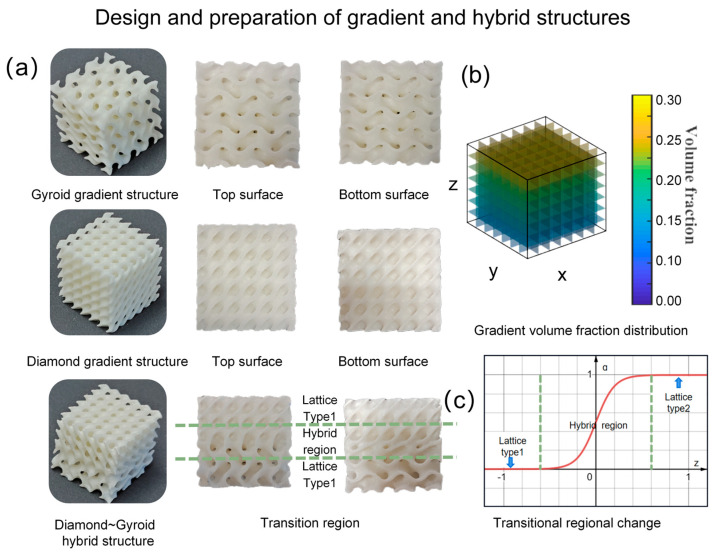
(**a**) Printed gradient and hybrid TPMS structure model. (**b**) Variations in the volume fraction of Gyroid and Diamond gradient structures with an average volume fraction of 0.2. (**c**) Structure change maps of hybrid TPMS were determined using the sigma function (SF).

**Figure 8 materials-16-07565-f008:**
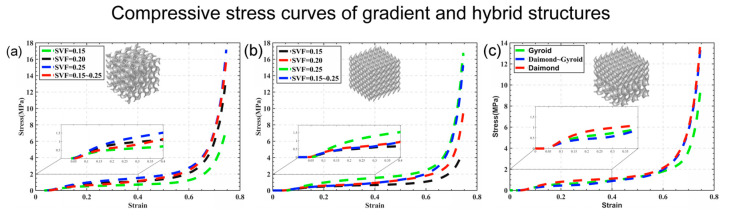
(**a**,**b**) Comparison of the stress–strain curves of gradient Gyroid and Diamond structures with homogeneous TPMS structures having a volume fraction of 0.15 to 0.25; (**c**) Comparison of the stress–strain curves of Gyroid and Diamond hybrid structures with homogeneous TPMS structures having the same volume fraction.

**Figure 9 materials-16-07565-f009:**
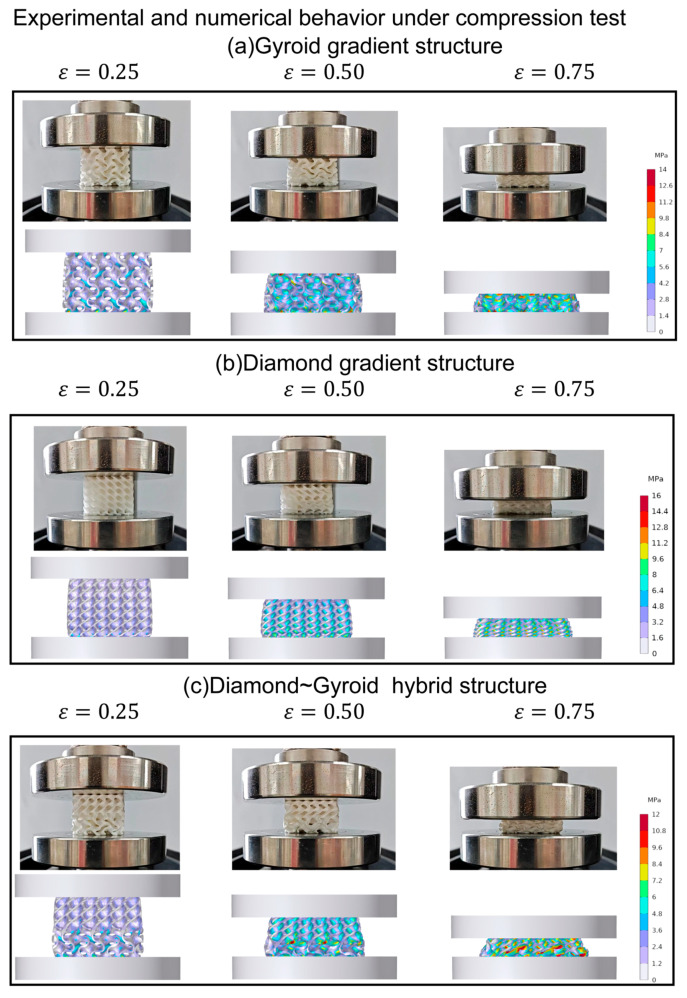
Compressive load finite element deformation analysis of gradient and hybrid TPMS structures.

**Table 1 materials-16-07565-t001:** SLS process parameters.

Process Parameter	Value
Laser power	20 W
Laser scan speed	2500 mm/s
Laser hatch spacing	0.1 mm
Powder deposition thickness	0.15 mm
Powder bed temperature	125 °C

**Table 2 materials-16-07565-t002:** Quality characterization of printed samples.

Type	Volume Fraction	Weight (g)	Dimension (X × Y × Z mm)
Uniform Gyroid structure	0.15	7.69	35.38 × 35.44 × 36.51
	0.20	10.65	35.33 × 35.55 × 36.50
	0.25	12.71	35.64 × 35.89 × 36.32
Uniform Diamond structure	0.15	6.95	35.32 × 35.22 × 36.12
	0.20	11.18	35.53 × 35.23 × 36.48
	0.25	13.51	35.25 × 35.64 × 36.33
Gradient structure	Average: 0.20	10.12	35.13 × 35.23 × 36.40
Gradient structure	Average: 0.20	11.17	35.17 × 35.88 × 36.65
Hybrid lattice structure	0.20	11.38	35.77 × 35.46 × 36.76

**Table 3 materials-16-07565-t003:** SLS parameters used in the production of lattice structures for mechanical testing.

Type	Solid Volume Fraction	$\mathrm{Elastic}\;\mathrm{Moulds}\;E\left(\mathrm{MPa}\right)$	$W\left(\mathrm{MJ}/m^{3}\right)$	$\Delta W\left(\mathrm{MJ}/m^{3}\right)$
Gyroid	15	2.28	0.64	0.23
Gyroid	20	3.03	1.03	0.37
Gyroid	25	3.44	1.6	0.59
Diamond	15	2.60	0.81	0.26
Diamond	20	3.25	1.35	0.44
Diamond	25	4.43	1.72	0.74

**Table 4 materials-16-07565-t004:** Mechanical property statistics of gradient/hybrid flexible TPMS structures.

Type (Volume: 36 × 36 × 36 mm^3^)	Solid Volume Fraction	Elastic Moulds E (MPa)	Energy Absorption $W\;\left(\mathrm{MJ}/m^{3}\right)$	Dissipation of Energy $\Delta W\;\left(\mathrm{MJ}/m^{3}\right)$
Gyroid (uniform)	20%	2.28	0.64	0.23
Diamond (uniform)	20%	3.25	1.35	0.44
Gyroid (gradient)	20%	2.66	1.26	0.43
Diamond (gradient)	20%	3.51	1.61	0.52
Gyroid~Diamond (hybrid)	20%	2.56	1.19	0.43

## Data Availability

Data are contained within the article.

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
