# Peer review of "The Laser Selective Sintering Controlled Forming of Flexible TPMS Structures"

_materials, 2023, doi:10.3390/ma16247565_

Round 1

Reviewer 1 Report (New Reviewer)

Comments and Suggestions for Authors

The paper focuses on the experimental and numerical analysis of three types (uniform, gradient and hybrid) structures based on Gyroid and Daimond Triply Periodic Minimal Surfaces lattices under compressive tests.

I see that this paper has already been corrected. However, it requires further improvements.

Comments:

1. line 191-195: add description (equations) of Neo-Hookean hyperelastic constitutive model.

2. line 198-200: explain in more detail where relationships 18 and 19 came from.

3. Figure 3: add a) and b)

4. explain in more detail where relationships 20 and 21 came from. From FEA?

5. Figure 4: Where in the text is the reference to this figure? Add a) and b). What means deltaE? Why you use V=30% in this figure?

6. Figure 5: hysteresis is hardly visible. Zoom in on the slice in the free area of each chart. Add a) and b). My question is: 0.75 strain was used in all three cycles? After first cycle specimen was not damaged?

7. Figure 6: add a) and b). What cycle represents this figure, first? Change figure caption, my proposition is: Experimental and numerical behaviour under compression test of Gyroid a) and Diamond b) lattice structures with a volume fraction of 0.20.

8. line 297: Was the gradient about the z-axis, or there was volume gradient? Explain it more detailed.

10. Figure 7 b): use legend in scope 0.15 to 0.25. Does this drawing contribute anything since the gradient was linear?

11. Figure 7 c): When you define in text sigma function (SF)?

12. Table 3: define deltaW

13. Lack of macro and micro photos of failed specimens. This is a very serious deficit. It absolutely must be corrected. Hence, there is a lack of explanation of the failure mechanisms of the lattice TPMS samples.

Comments on the Quality of English Language

1. sentences that are too long are often used

Author Response

Thank you for your careful review, please see word for details. This is my first submission and I will try my best to meet the requirements of the journal. Wish you a happy life.

Reviewer 2 Report (New Reviewer)

Comments and Suggestions for Authors

The manuscript entitled “Laser selective sintering controlled forming TPMS structure” focus on the modelling of equations able to describe triply periodic minimal surface structure (TPMS). These equations are able to evaluate the mechanical properties as well as the energy adsorption and deformation behavior. The use of laser selective sintering allowed to build the designed TPMS and validate the model by performing compression tests. The agreement of numerical and experimental results is within the acceptable limit of 15%. These results put the basis to implement new hybrid architecture of triply periodic minimal surface lactice structure.

The topic deals with recent developments in the field of Metamaterials. The manuscript can be accepted if the following comments are addressed.

Materials and methods

-     -The authors could clarify how the process parameters listed in Table 1 have been chose;

-    -The authors could clarify how many specimen have been used to perform compression tests.

Results

-        The authors pointed out the relevant importance of the volume fraction on the extracted mechnanical properties, deformation behavior and energy developed during testing. So stated, the authors could clarify if the aspect ratio of a triply periodic minimal surface lactice structure would potentially affect the above mentioned properties.

Additional suggestions

The following suggestions would help to enhance the quality of the manuscript:

-      -Graphs in Figure 2 a) and in Figure 8 should be enhanced. The titles of the axis and the legends are not clearly readable.

-    -Captions should be placed in the same page of the Figures/Tables they belong to (i.e. Figure 6);

-         -Table 2 should be placed where it is first mentioned in the main test;

-        -Spacing is required after a coma or a full stop (i.e. line 210). Please correct all  these missing in the manuscript;

-     -The unit of a measure should be written with capital letter (see in Table 1 where 20 W should be used instead of 20 w).

-        Please correct some language typos (for example, Diamond should be used instead of Daimond).

Author Response

Thank you for your careful review, please see word for details. This is my first submission and I will try my best to meet the requirements of the journal. Wish you a happy life.

Round 2

Reviewer 1 Report (New Reviewer)

Comments and Suggestions for Authors

All my comments have been taken into account.

This manuscript is a resubmission of an earlier submission. The following is a list of the peer review reports and author responses from that submission.

Round 1

Reviewer 1 Report

Comments and Suggestions for Authors

This manuscript discusses the controllable fabrication of two flexible TPMS structures using selective laser sintering. The purpose of this paper is to develop lattice structures with controllable and adjustable mechanical properties using Selective Laser Sintering (SLS) and uniform TPMS structures. Using thermoplastic polyurethane (TPU) material, we model uniform, gradient, and hybrid lattice structures. By using compression experiments and finite element analysis, they investigate the mechanical performance, deformation behavior, energy absorption characteristics, and shock resistance of the designed TPMS lattice structures. Based on the results of our parametric modeling approach, we have demonstrated that flexible TPMS lattice structures with the desired mechanical properties can be fabricated effectively. In addition to enhancing elastic modulus, energy absorption, and shock resistance, our gradient, and hybrid TPMS lattice structures significantly outperform uniform TPMS lattice structures.

Minor corrections need to be made before publication:

1. There are many grammatical errors in the manuscript, and the author needs to correct them.

2. This paper's abstract needs to be improved since it seems to contain a lot of unnecessary discussion.

3. The quality of Figure 8 needs to be improved.

4. The author should improve the discussion and prepare a table showing the advantages of his or her claims.

Comments on the Quality of English Language

Need to improve.

Reviewer 2 Report

Comments and Suggestions for Authors

It is a paper that is well written, it tackles the aim of the paper very smoothly and therefore, I do not have many comments about the content.

Despite that, I would give different bullet points that you could carry to improve the paper:

1) Analysis of the 3D shape fidelity using a uCT or a scanner

2) Chemical and physical analysis of the material used

3) Looking for a possible application, for instance, in medicine, by carrying out cell-assays

4) Add more information in the introduction about the AM technologies, maybe why choosing SLS and not other, etc.

Author Response

The following is our response to your insightful comments, which we greatly appreciate.
1. As you said, the geometry of the lattice structure has a crucial influence on the mechanical properties. Existing SLS preparation technology for the tpu sample exhibits an apparent phenomenon of contraction; therefore, the quality of the prepared sample was described in the experimental method description. The surface roughness of ct will be optimized following the resolution of the shrinkage problem.
2. the properties of materials indeed have a significant effect on the performance and process parameters of the prepared samples; this is also the topic of the second research paper that we have been diligently producing. In this investigation, we are interested in the hyperelastic response of the structure, so we select the equipment manufacturer's recommended printing parameters.
In response to your suggestions, the following modifications have been made: We begin by extending the conclusion, summarizing the impact of the research results on the application of sports equipment, and citing relevant literature to demonstrate the significance of the research.
4. Your comments are excellent, and we have amended the introduction to reflect the current research status of SLS printing technology.

Reviewer 3 Report

Comments and Suggestions for Authors

In the manuscript titled "Selective Laser Sintering for Controllable Fabrication of Flexible TPMS Structures,” the authors propose a parametric approach based on selective laser sintering (SLS) and uniform TPMS (three-period minimal surface ) structures to develop lattice structures with controllable and tunable mechanical properties.
Although the topic is interesting and relevant to the field, due to the fact that 3D printing technology is widely used nowadays and has great potential for various industries, I could not find any novelty in this work. Commercially available thermoplastic polyurethane (TPU) was used to model various structures. Similar research has already been published in https://doi.org/10.3390%2Fmi13071017.
The manuscript is well organised, the authors have applied the scientific methods and they are adequately described. The results are clearly presented and reproducible due to the detailed description of the methods. In the results, the authors present the findings on the properties and mechanical performance of the materials studied and compared to the TPU metamaterial structures used.
Although the research is interesting, an additional scientific contribution should be made by presenting a summary of the results that gives a better insight into the possibilities of using presented model under the observed conditions. In the introduction it is written that the observed results could provide a new method for the design of flexible metamaterials in sports equipment, but the results and the discussion did not show any correspondence between the obtained results and the possible use in the production of sports equipment.
It can be said that a summary of the presented method should be elaborated on the observed properties of the materials in terms of influence on the performance of sports equipment, which would achieve a high scientific soundness of the research with possible recommendations for application.
In my opinion, the manuscript can be published with these additional changes.

Author Response

We appreciate the valuable advice you have provided. Below is a summary of our responses to your feedback and the modifications we have made.
1. According to your observation, both parametric lattice structure molding and commercial thermoplastic polyurethanes have been extensively investigated. However, there still needs to be more research between the two fields. How to accurately test and characterize the hyperelastic response of commercial thermoplastic polyurethanes; How to design the structure of a parametric flexible lattice. In order to assist future researchers in characterizing and testing flexible lattice structures more reasonably, we have reviewed a vast amount of literature in these disciplines and summarized their methodologies in this manuscript. In response to this issue, we revised the abstract, introduction, discussion, and conclusion to clarify the research's significance.
2. You have pointed out the logical flaws in our manuscript regarding the relationship between the application of sports equipment and the research in a reasonable manner. We have meticulously revised the abstract and introduction to emphasize that the focus of our research is the parametric design of flexible lattice structure and hyperelastic response, and we have removed the description of the applicability to sports equipment from the abstract. Concurrently, the conclusion is divided into two sections: the discussion and the conclusion. In the discussion, the research's outcomes and motivations are elaborated. In conclusion, the influence of these studies on the field of athletic apparatus is discussed.
3. Finally, we re-proofread and translated the manuscript word-for-word, simplified the full text, and added more references to support the research context, objects, methods, and conclusions to satisfy the journal's requirements.

Reviewer 4 Report

Comments and Suggestions for Authors

The paper focuses on the fabrication of flexible TPU metamaterial structure by SLS. Authors realized the mathematical modeling, simulation, and characterization of the designed structures by the testing of their mechanical properties. I do have a few comments and suggestions:

1, Please do not introduce abbreviations in Abstract. In addition, the Abstract does not adequately summarize the paper's contents. I recommend including additional information in the Abstract and expanding the Abstract with a major result.

2, FCC, BCC, and SHC are not introduced abbreviations.

3, Line 71 and 72: Authors state the sentence: “Gradient and hybrid TPMS lattice 71 structures are currently hot research topics [11, 19]”. The literature 19 is from 2014. This is not currently hot research topic.

4, Equations 2 and 3 are not sufficiently described.

5, The title of the Figure 1 is not correct. It begins with “depicts”.

6, How was the parameters of the designed TPMS structures chosen (lines 150-157)?

7, The mechanical response curve of the regular lattice structure in the Fig. 2a is based on the measured values or it is only the general response curve? If it is the general response curve, please state some literature.

8, Lines 217-219 are probably from the template.

9, The Conclusion is too long. I recommend dividing it to Discussion and Conclusion.

10, 25 references (21 in Introduction) are insufficient for this kind of paper.

Based on the comments mentioned above, I suggest a major revision of the paper.

Reviewer 5 Report

Comments and Suggestions for Authors

The authors tried to fabricate a flexible TPU metamaterial structure by SLS, and to realize the parametric design method and mechanical property characterization with controllable and adjustable mechanical properties. Designing the 3D printing parameters for flexible polymer processing is very important from both academic and industrial viewpoints. The results the authors supplied are also novel and interesting. Therefore I would recommend this paper for publication. 

Author Response

Thank you for your review. In an effort to meet the journal's requirements, we revised and translated the manuscript word-for-word, streamlined the full text, and added more references to support the research context, objects, methods, and conclusions.

Round 2

Reviewer 3 Report

Comments and Suggestions for Authors

Authors improved the manuscript and it can be accepted in present form.

Author Response

Thank you for your comments. At present, the academic editor has rejected the article and suggested us to revise it and re-submit it. We will revise it according to your comments and hope to meet your requirements when re-submitting it.

Reviewer 4 Report

Comments and Suggestions for Authors

The paper has been much improved. I recommend to accept the paper in the current form.

Author Response

(The authors gave the same response as above.)
